# Healthy Behavior for Preventing Cognitive Disability in Older Persons

**DOI:** 10.3390/ijerph22020262

**Published:** 2025-02-12

**Authors:** Fulvio Lauretani, Antonio Marcato, Crescenzo Testa

**Affiliations:** 1Geriatric Clinic Unit, Medical-Geriatric-Rehabilitation Department, University of Parma, 43121 Parma, Italy; 2Department of Medicine and Surgery, University of Parma, 43121 Parma, Italy; crescenzo.testa@unipr.it; 3Rehabilitation Unit, Medical-Geriatric-Rehabilitation Department, University Hospital, 43126 Parma, Italy; amarcato@ao.pr.it

**Keywords:** dementia, older persons, physical activity, sedentary habits

## Abstract

Sufficient levels of physical activity are fundamental for preventing cardiovascular disease, dementia, and ultimately disability in older persons, yet this protective factor is nullified when excessive hours are spent in continuous sitting. Balancing physical activity and sedentary behavior is crucial for influencing metabolic parameters and vascular patterns, both central and peripheral, thereby reducing the risk of cardiovascular diseases, vascular dementia, and cognitive impairment. The primary goal of geriatric medicine is to improve quality of life and prevent disability by promptly identifying frail older individuals, thus mitigating both cognitive and motor impairments. Achieving this objective requires not only the optimization of pharmacological treatments but also the active promotion of a healthy lifestyle. In this context, investigating preclinical stages of disability, such as Motoric Cognitive Risk (MCR) Syndrome, which integrates physical and cognitive components of decline, becomes essential. However, despite robust evidence supporting these interventions, greater efforts are needed from the geriatric medical community to bridge the gap between scientific recommendations and everyday clinical practice. Integrating these guidelines into routine care is pivotal for delivering personalized interventions that address both physical inactivity and prolonged sedentary behavior. More research should aim to strengthen this balance, providing clearer, actionable strategies for clinicians to implement, thereby fostering the formation of evidence-based public health guidelines on physical activity specifically tailored for older persons.

## 1. Introduction

Physical activity in older persons seems to be the strongest protective factor for preventing dementia and mortality [1]. A recent study showed that the risk of mortality among adults aged 60 years and older progressively decreased when the number of steps increased to 6000–8000 per day. The same results were achieved among adults younger than 60 years when increasing their number of steps to 8000–10,000 per day [2]. This number of daily steps is also relevant to the cerebral deposition of β-amiloid [3]. More specifically, there was a significant interaction between physical activity and Aβ burden, such that greater physical activity was associated with slower Aβ-related cognitive decline and volume loss. Lower and higher Aβ burden groups (2900 steps per day and 8300 steps per day, respectively) were created using the median Aβ levels in Aβ-negative and Aβ-positive groups, which correspond to a distribution volume ratio value of 1.1 and 1.9. This type of association is not surprising given that high fitness and dementia have been longitudinally associated.

Among Swedish women, for example, high cardiovascular fitness (CRF) in midlife was associated with a decreased risk of subsequent dementia [4]. These authors wrote that promotion of a higher rate of cardiovascular fitness may be included in strategies to mitigate or prevent dementia. Physical activity, in general, has positive effects on dementia risk but lifelong, high levels of cardiovascular fitness are the most effective. Recently, these findings were confirmed by a robust study, where higher cardiorespiratory fitness corresponded to higher cerebral myelination in aging. The authors concluded that higher VO2_max_ is associated with greater cerebral myelination, particularly in middle-aged and older adults, providing insights into the potential protective role of CRF in attenuating demyelination in aging [5].

Even a dose–response association has been reported between levels of physical activity and mortality due to Alzheimer’s disease [6]. These authors found that a minimal amount (i.e., 50% of the optimal amount) at 40 min/week and an optimal amount (i.e., the nadir of the curve) at 140 min/week led to reduced Alzheimer’s disease-related mortality.

For the USA, they estimated that 40 weekly min of vigorous physical activity (VPA, defined as at least 20 min that causes heavy sweating or large increases in breathing or heart rate) would prevent 12.238 deaths per year and 140 weekly min of VPA would prevent 37.710 deaths per year, compared with a scenario in which US adults did not do any VPA [6].

However, it is not only levels of physical activity that are relevant for preventing dementia. In addition, there is also a strong correlation between sedentary behavior and a sharp increase in the occurrence of dementia [7]. Indeed, these authors emphasized that among older adults, the more time older adults spent in sedentary behavior, the greater the incidence of all-cause dementia.

The balance between levels of physical activity and hours in a sedentary position was also reported in a complete study. This study showed that the beneficial net effect of physical activity was nullified when a person spent more than 10 h a day seated in a chair [8].

This leads us to our guiding question: should we focus on the total daily amount of physical activity or balance between levels of physical activity and sedentary behavior?

## 2. Should We Focus Our Attention More on Mitigating a Sedentary Lifestyle Rather than Trying to Increase Physical Activity in the Elderly?

Until recently, a primary focus of physical activity guidelines within public health practice has been to recommend moderate-to-vigorous-intensity physical activity (MVPA). Most research on physical activity and brain health focuses on MVPA. However, for most older adults, only a small portion of the day is spent in moderate-to-vigorous physical activity (MVPA).

However, emerging evidence suggests that replacing time spent in sedentary behavior with light-intensity physical activity can improve metabolic control [9]. In detail, sedentary behavior, defined as too much sitting, particularly uninterrupted sitting, as a distinct concept from levels of physical activity, has been shown through longitudinal findings to adversely affect cardiovascular health. Prolonged, uninterrupted sitting detrimentally affects several biological processes related to cardiovascular risk and spending many hours of sitting annuls the total physically active time, negating the cardiovascular benefits of skeletal muscle activity.

A study conducted among university employees in Ethiopia showed that for each increase of one hour per day of sitting there was a significant increase in body mass index (BMI), fasting blood glucose, diastolic and systolic blood pressure, and waist circumference after adjustment for all covariates [10]. A sophisticated recent study suggested that sitting for many hours uninterrupted influences the endothelium, particularly of the lower leg, and that prolonged constriction of resistance arteries can lead to modifications in the structural characteristics of the vascular wall, with micro- and macrovascular changes stiffening arteries due to impaired vasorelaxation responses [11].

Much less is known about the implications of this increase in light physical activity for brain health. Added to this lack of research, we have additional research that suggests that even emphasizing the importance of light physical activity is insufficient in motivating elderly patients to change the amount of their day spent in a chair [12] (Figure 1).

Moreover, from a clinical and public health perspective, prescribing moderate-to-vigorous physical activity targets is often impractical for many older adults, particularly those who are frail or have multiple comorbidities. Because they may be unable to achieve or maintain higher intensity levels, emphasizing light-intensity physical activities and regular breaks from prolonged sitting are both more feasible and more beneficial for this population. Physical activity levels are lowest among older adults (aged > 65 years), who are also at the highest risk of cardiovascular disease compared with other age groups. This is particularly concerning as sedentary behaviors, such as prolonged sitting or reclining, comprise a large portion of many people’s waking hours worldwide.

Sedentary behavior that typically involves long periods of sitting during waking hours might have physiological consequences that are distinct from those of periods of a lack of moderate–vigorous-intensity physical activity, often referred to as exercise. Unfortunately, in contrast to the well-defined physical activity guidelines, sedentary behavior guidelines remain vague and non-specific, providing little guidance as to what might be considered to be the total “safe” sitting duration per instance or per day.

The only tentative attempt to define sedentariness is that of the World Health Organization (WHO), which has described the sedentary lifestyle as a behavior characterized by very low energy expenditure, generally less than 1.5 MET (Metabolic Equivalent of Task), associated with activities such as sitting, watching television, working on the computer, or using other electronic devices.

Interestingly, sedentary time, independent of time spent being physically active, has been identified as a modifiable risk factor for mortality [8]. Hence, it is not surprising that sitting has been called “the new smoking”, which refers to its role as a substantial risk factor for cardiovascular disease.

Over the past few decades, observational studies of physical activity have relied on self-reporting, making it difficult to value the real measure of activity. More precise measurements can now come from small wearable accelerometers that can provide data across the entire day. Noteworthy studies that have used accelerometers on young (mean age: 22 years) and old (mean age: 65 years) people have shown that sedentary time is inversely associated with the measure of cardiorespiratory fitness, even after the person has been involved in moderate–vigorous-intensity activity [13].

Many older adults spend most of their waking hours sitting. And this pattern is further amplified and reinforced by social isolation. Physical inactivity, defined as a level of activity that is insufficient to meet current physical activity guidelines, has long been known to be a major contributor to the risk of cardiovascular disease. Insufficient physical activity has also long been recognized as a risk factor for major chronic diseases and mortality.

Trends in Adherence to the Physical Activity Guidelines for Americans for Aerobic Activity and Time Spent on Sedentary Behavior among US Adults, 2007 to 2016, have observed a significant increase in time spent on sedentary behavior during the past 10 years. So, sedentary behavior (put simply, too much sitting) has adverse health consequences that are distinct from those arising from too little exercise [14]. And, more specifically, sedentary time characterized by a sitting posture in the absence of skeletal contractile activity has increased on average by around 1 h per day in the last several years [15]. Sedentary behaviors—primarily watching television or talking on or interacting with a cellular phone—occupy the largest proportion of adults’ waking hours.

## 3. Can Prolonged Sitting Posture Interfere with Mental Health?

Higher levels of sedentary behavior are also associated with increased risk for type 2 diabetes, cardiovascular disease, some cancers, and an all-cause of mortality. Evidence linking sitting to poor mental health and dementia is scarce and much remains to be understood. To date, most of the experimental and observational studies have focused on metabolic risk. Stronger evidence is needed to discover whether there is an underlying biological mechanism leading to deleterious health consequences on the brain. Preliminary evidence suggests, however, that glycemic variability may influence brain health and cognition [9]. A close relationship between cerebral vascular changes and atrophy and cognition exists and a recent study emphasized that white matter (WM) injures are associated with cerebral atrophy and dementia [16]. The evidence seems increasingly clear that a triad of vascular impairment, cerebral atrophy, and cognitive decline represents critical age-related conditions that significantly impact health [17,18,19,20,21]. However, the specific mechanisms through which vascular damage influences amyloid deposition remain largely unknown.

Emerging evidence indicates that nearly 40% of dementia cases can be attributed to modifiable risk factors, including inadequate sleep and harmful alcohol consumption. Poor sleep quality disrupts metabolic and inflammatory processes in the brain, potentially accelerating the deposition of amyloids and hastening the progression toward cognitive impairment. Likewise, excessive alcohol use can exert neurotoxic effects, contributing to vascular damage, brain atrophy, altered neurotransmitter function, and, in some studies, increased amyloid accumulation. Addressing these factors—by improving sleep hygiene and limiting or avoiding alcohol intake—presents a substantial opportunity to reduce the incidence of dementia. By targeting these modifiable behaviors in middle and older adulthood, individuals and healthcare systems alike can work toward delaying or preventing cognitive decline in a significant portion of at-risk populations [21].

Based on data from nearly 50,000 adults in the United Kingdom, NIA-funded researchers have shown an association between dementia risk and daily sedentary behavior. Though the study cannot establish a causal link, it does support the idea that more time spent not moving—such as sitting while watching TV, working on a computer, or driving—may be a risk factor for dementia [18]. Accelerometer data indicated that older adults who spent an average of 10 h per day in sedentary behaviors had a higher likelihood of developing all-cause dementia [7]. Importantly, it is becoming clear that environmental-, social-, and individual-level factors, which can positively or negatively influence how much time is spent in sedentary behavior, are distinct from factors linked to the adoption and maintenance of exercise.

Laboratory experimental evidence linking sedentary behavior with biomarkers of chronic disease risk (i.e., 2 h fasting glucose) also emerged [20], but more findings are required to establish the exact mechanism through which sedentariness is linked to cognitive decline in older persons.

Bed rest studies have clearly demonstrated that prolonged, unbroken sitting is harmful; however, the sophistication of measures needed to assess sedentary accumulation patterns in free-living adults means that this research area has only just begun, especially since it is not easy to monitor people during their waking hours.

Table 1 summarizes the potentially preventable risk factors for dementia [21].

## 4. Where Body Meets Brain: A Shared Path of Decline?

For many years, the clinical approach to older patients has often been shaped by a dichotomous viewpoint, wherein motor dysfunction and cognitive impairment were evaluated as separate entities. Geriatric assessment units have traditionally divided their focus: one part is dedicated to cognitive complaints and another to motor disturbances. However, emerging evidence has questioned this “traditional” separation, emphasizing the need for a more integrated viewpoint on aging-related conditions. A key turning point in this paradigmatic shift was the work of Joe Verghese and colleagues, who introduced the concept of Motoric Cognitive Risk (MCR) Syndrome as a valuable clinical tool [20]. MCR aims to overcome the motor/cognitive dichotomy by recognizing how slowed gait and subjective cognitive complaints frequently coexist and may signal a common physiopathological mechanism. In other words, rather than evaluating motor and cognitive impairments in isolation, MCR advocates for a unified approach that may foster earlier and more precise risk stratification for dementia and disability. Given the dramatic projections for the global increase in dementia by 2050, such innovation is desperately needed. According to recent forecasts, the worldwide prevalence of people living with dementia is expected to rise from about 57.4 million in 2019 to 152.8 million by 2050. This substantial increase demands updated frameworks for prevention and clinical care—especially to detect those “at-risk” individuals earlier and address modifiable factors that may delay or halt the progression toward overt dementia [21].

First introduced in 2013, MCR combines two simple clinical indicators:Slowed gait speed, measured objectively (e.g., a timed 4 m walk).Subjective cognitive complaints, self-reported by the patient (e.g., perceived memory difficulties or other cognitive issues).

These criteria must be present in an individual who has not yet been diagnosed with dementia or any significant functional limitation, so as to capture the prodromal stage of neurodegeneration [22]. Although different studies have used slightly varying diagnostic criteria, the average worldwide prevalence of MCR is estimated at around 10% in older adults. Critically, older adults who meet the MCR criteria face an approximately threefold higher risk of converting to overt dementia [23].

The relevance of MCR lies in bridging the gap between motor and cognitive dysfunction, allowing clinicians to screen for dementia risk using simple and non-invasive methods. Where once these domains were viewed independently, it is now clear that shared neuropathological factors, such as small-vessel disease, white matter lesions, inflammation, and neurochemical imbalances, can simultaneously compromise gait regulation and cognitive processes [24,25]. This means that slowed gait speed can be an early alarm for more extensive neural changes, including those that eventually manifest as dementia.

Early identification of MCR is particularly pressing when we consider the interplay of depressive symptoms and social isolation in dementia risk [26]. For instance, lack of social engagement and depression have been shown to be potent factors influencing the cognitive trajectory of older adults. These psychosocial elements may be linked to neurobiological circuits, such as the reward system, which can influence mood, motivation, and, in turn, cognitive performance [24,27]. When depression and a blunted reward system coincide in older adults who already exhibit slowed gait and subjective cognitive complaints, the overall risk of cognitive decline may be amplified. Conceptualizing a loop among the reward system, depressive mood, and declining cognition highlights the synergistic nature of these processes, suggesting that targeted interventions in any one domain (e.g., improving mood, promoting physical activity, enhancing sociability) might have cross-benefits in others [27].

Growing evidence suggests that the pathophysiological mechanisms underlying MCR may include the following:-Vascular dysregulation: chronic sedentary behavior and other cardiovascular risk factors may compromise cerebral blood flow, contributing to both slowed gait and early cognitive difficulties.-Microstructural white matter changes: subcortical and periventricular lesions can affect frontal–subcortical circuits, which are essential for both executive function (cognitive) and gait regulation (motor).-Neurotransmitter imbalances: dopamine, acetylcholine, and other neuromodulators are deeply involved in motor planning and cognitive processing, linking changes in gait speed to subtle cognitive deficits.

From a public health standpoint, recognizing these overlapping pathways underscores the importance of modifiable risk factors. Depression, physical inactivity, and social isolation are all factors that, if detected early, can be addressed through comprehensive geriatric interventions:-Exercise and physical therapy to improve muscle strength, gait stability, and cardiovascular health.-Cognitive stimulation or dual-task training (performing motor and cognitive tasks simultaneously) to strengthen neural circuitry.-Psychological support and screening for depression, ensuring timely referral to mental health professionals.-Lifestyle interventions that combat prolonged sitting, encourage frequent breaks, and promote meaningful social interactions.

The prevalence and predictive value of MCR call for routine clinical assessment of older individuals who report subjective cognitive complaints and exhibit slowed gait. This straightforward combination can flag a population at heightened risk for progressing to dementia. As the data on MCR accumulate, future research should focus on the following:-Refining diagnostic criteria to ensure consistent and comparable findings across diverse populations.-Exploring targeted interventions (e.g., structured exercise programs, cognitive remediation, depression screening) to determine whether addressing MCR early can prevent or delay dementia onset.-Longitudinal studies that integrate brain imaging, neuropsychological assessments, and wearable technologies (for gait speed and physical activity monitoring) to unravel the precise physiopathological substrates of MCR.

## 5. Fitness = Social Engagement: The Synergy for Healthy Aging

Emerging evidence highlights the profound connection between physical activity (PA), sedentary behavior, and social engagement in shaping the cognitive, cardiovascular, and overall well-being of older adults. According to the Global Consensus on Optimal Exercise Recommendations for Enhancing Healthy Longevity in Older Adults, while sufficient levels of PA significantly reduce the risks of dementia, cardiovascular diseases, and disability, these benefits are negated by prolonged sedentary behavior [28]. Extended sitting, often exceeding 10 h daily, impairs endothelial function, diminishes vascular reactivity, and exacerbates metabolic dysregulation, counteracting the protective effects of skeletal muscle activity.

However, the consensus emphasizes that even light-intensity PA and frequent breaks from sitting can restore metabolic balance and improve vascular health, particularly in populations prone to motor and cognitive decline.

Beyond the physical domain, social interaction emerges as a pivotal determinant of healthy aging. Social isolation, often a byproduct of aging, is recognized as a modifiable risk factor for depression, cognitive decline, and the development of Motoric Cognitive Risk (MCR) Syndrome. Dysfunction in the dopaminergic reward system, central to regulating responses to both social stimuli and physical rewards, has been associated with apathy, depression, and accelerated neurodegeneration. Research demonstrates that maintaining social engagement through group participation, volunteering, or family connections activates the reward system, enhances motivation, and alleviates depressive symptoms [26]. These findings strongly align with the consensus’ recommendation to incorporate socially engaging activities into exercise programs, amplifying both psychological and physiological benefits.

The dynamic relationship between social engagement and physical activity is particularly critical in combating frailty. Socially active older adults are more likely to adhere to exercise regimens, reduce sedentary behavior, and bolster motor-cognitive resilience. Similarly, physical activity fosters social participation by improving mobility, alleviating depressive symptoms, and enhancing self-confidence. This interplay creates a virtuous cycle in which physical activity and socialization jointly protect against cognitive and functional decline, reinforcing the consensus’ call for multicomponent interventions that integrate exercise with social engagement as essential components of geriatric care [29].

By combining physical activity with strategies to minimize sedentary behavior and encourage social interaction, clinicians can adopt a comprehensive approach to mitigating the risks associated with aging. Future public health guidelines should embrace this holistic perspective, emphasizing the synergy between PA and social engagement to strengthen resilience against motoric and cognitive impairments while enhancing the quality of life in older populations.

## 6. Physical Activity: How Much Is Enough for Healthy Aging?

In Table 2, the most recent recommendations on exercise guidelines for older adults are summarized.

These evidence-based recommendations, which align with the World Health Organization’s guidelines, highlight the importance of a multicomponent approach [28]. Combining aerobic, strength, balance, and flexibility exercises while minimizing sedentary behavior supports not only physical health but also cognitive and emotional well-being, making them essential for promoting healthy aging.

## 7. Conclusions

Addressing Motoric Cognitive Risk Syndrome provides a comprehensive framework for the prevention and intervention of age-related cognitive and motor decline. By focusing on modifiable factors such as depression, social isolation, and lifestyle-related risks, geriatric care can effectively target the early stages of neurodegeneration. Proactively managing MCR has the potential to flatten the steep trajectory toward cognitive decline, thereby reducing disability and improving the quality of life for the aging global population.

Sufficient levels of physical activity are fundamental for preventing dementia, cardiovascular diseases, and ultimately disability. However, the protective benefits of physical activity are nullified when older adults spend excessive hours in a sedentary position, such as prolonged sitting (Figure 2). Balancing physical activity and sedentary behavior is critical to optimizing metabolic parameters and vascular health, affecting both central and peripheral arteries. This balance plays a pivotal role in reducing the risks associated with vascular dementia, cerebral atrophy, and cognitive impairment, while also mitigating the risk of motoric and cognitive disabilities.

The primary goal of geriatric medicine is to enhance quality of life and prevent disability by identifying frail individuals and addressing their unique needs. This requires going beyond pharmacological interventions to actively promote a healthy lifestyle, integrating evidence-based physical activity regimens into routine clinical practice. Programs should incorporate aerobic, strength, balance, and flexibility exercises tailored to individual capabilities while minimizing sedentary behavior.

Future public health strategies should focus on bridging the gap between scientific recommendations and everyday clinical practice. Providing clinicians with clear, actionable guidelines for balancing physical activity and sedentary behavior is essential to achieving meaningful improvements in health outcomes for older adults. Furthermore, fostering a holistic approach that includes promoting social engagement alongside physical activity can create a synergistic impact on both cognitive and physical health.

In conclusion, embracing a multidisciplinary and proactive approach to MCR and lifestyle modification offers the greatest promise for mitigating the challenges of aging, empowering older adults to maintain independence and enhancing their overall well-being.

## Figures and Tables

**Figure 1 ijerph-22-00262-f001:**
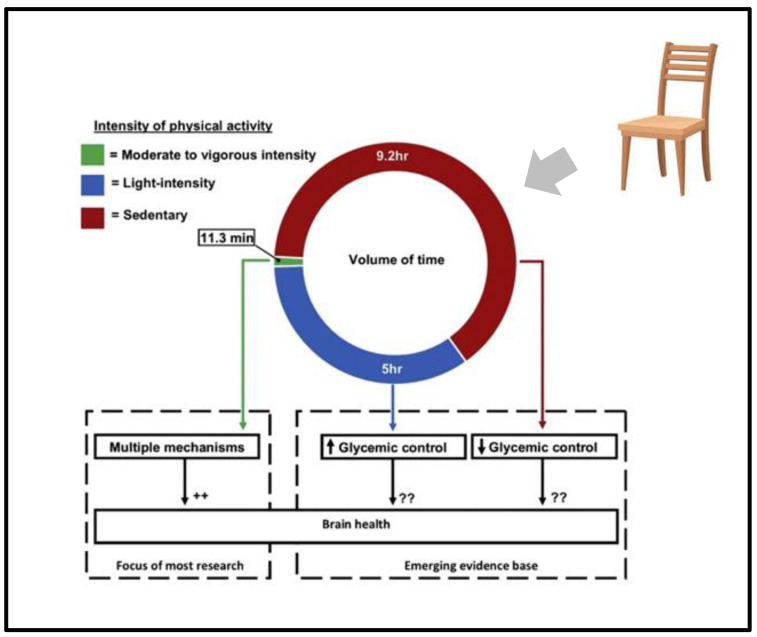
Intensity of physical activity and its effect on brain health. ++ means: many papers indentified different mechanism; ?? uncertain mechanism proposed; arrow means: Many papers indicates that less more 5 h of activity produce worsening of glicemic control, more than 9.2 h of activity produce improvement of glicemi control.

**Figure 2 ijerph-22-00262-f002:**
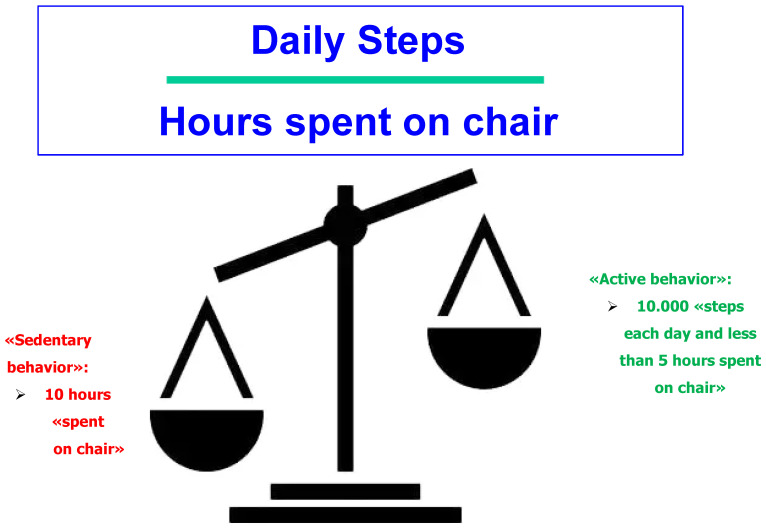
Healthy behavior for preventing cognitive decline in older persons.

**Table 1 ijerph-22-00262-t001:** Types of activities that could produce cognitive improvement.

Facilitator	Description/Rationale	Practical Examples/Interventions
Regular Physical Activity	Improves cardiovascular health, reduces risk of chronic diseases (e.g., diabetes, hypertension), and enhances cognition and mood.	- Daily walks, chair exercises, or group fitness classes - Light-intensity activities to break up sitting
Reduced Sedentary Behavior	Prolonged sitting can negate the benefits of exercise by compromising metabolic control and vascular health.	- Standing or light movements every 30 min - Tracking sedentary hours with wearable devices
Social Engagement	Enhances mood, reduces risk of depression, and supports cognitive reserve by stimulating social and mental activities.	- Community groups, volunteering, social clubs - Regular interaction with family and friends
Educational Support	Increases awareness of healthy lifestyle habits and empowers older adults and caregivers to manage chronic conditions effectively.	- Workshops on nutrition; medication management - Support groups and caregiver education sessions
Multidisciplinary Teamwork	Integrates expertise from geriatricians, nurses, physical therapists, nutritionists, and social workers for holistic care.	- Comprehensive geriatric assessments - Interdisciplinary case conferences and care plans
Environmental Modifications	Ensures safety, fosters independence, and supports aging in place through design tailored to older adults’ physical limitations.	- Handrails, slip-resistant floors, well-lit hallways - Universal design principles in homes and public areas
Healthy Diet and Sleep	A nutrient-rich, plant-based diet supports cardiovascular and cognitive health; adequate sleep promotes cognitive function.	- Emphasizing vegetables, fruits, whole grains - Sleep hygiene education and night-time routines
Avoidance of Risky Substances	Minimizes harm linked to alcohol or tobacco use, reducing the risk of cardiovascular disease, cancer, and dementia.	- Smoking cessation programs - Screening for alcohol misuse and offering counseling
Psychological Well-Being	Addressing mood disorders and stress can help prevent isolation and enhance overall quality of life and cognitive health.	- Regular screening for depression - Referral to mental health services if needed

**Table 2 ijerph-22-00262-t002:** Schematic types and amount of physical activities for improving frailty.

Activity Component	Frequency and Duration	Intensity/Notes	Examples
Aerobic Exercise	Total of 150–300 min/week (moderate intensity) OR 75–150 min/week (vigorous intensity)	- Moderate: brisk walking, water aerobics - Vigorous: jogging, cycling	- Total of 30 min of brisk walking 5 days/week - Total of 25 min of jogging 3 days/week
Strength Training	At least 2 days/week	- Focus on major muscle groups (arms, legs, core)	- Resistance bands - Light weights or bodyweight exercises
Balance and Fall Prevention	At least 3 days/week if at risk of falls	- Improves stability and prevents injuries	- Tai Chi, yoga - Balance-specific routines (e.g., standing on one foot)
Flexibility	Incorporate stretching sessions regularly (e.g., daily or on exercise days)	- Enhances range of motion and joint health	- Gentle stretches targeting calves, hamstrings, shoulders
Reducing Sedentary Time	Break sitting every 30 min with light activities or standing	- Short bouts of movement can significantly improve metabolic outcomes	- Standing breaks while watching TV - Brief household tasks (tidying, light chores)
Adaptations for Frailty	Begin with low-intensity, short-duration activities; progress gradually as far as can be tolerated	- Individualized approach based on functional status and comorbidities	- Seated exercises for very frail individuals - Supervised programs in a clinical setting

Notes: Moderate intensity activities typically allow for conversation but not singing. Vigorous-intensity activities make speaking in full sentences difficult. Frail older adults or those with complex comorbidities may need further customization and medical clearance prior to starting.

## Data Availability

No new data were created or analyzed in this study.

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
