# Peer review of "Healthy Behavior for Preventing Cognitive Disability in Older Persons"

_ijerph, 2025, doi:10.3390/ijerph22020262_

Round 1
Reviewer 1 Report
Comments and Suggestions for Authors
Authors have written an interesting article about very important topic.
Few little things need to be corrected;
Lines 81-83: However, or most elderly patients, only a very small proportion of the day is spent on activities that would be characterized as MVP.
Please clarify this sentence.
Lines 107-108: Moreover, from a clinical and public health perspective, identifying a target of moderately vigorous intensity would be unrealistic for this population.”
Please clarify this idea/sentence.
Lines 162-163:
”…but that clear mechanisms by vascular damage influence amyloid cerebral deposition is largely unclear.”
Please clarify this idea/sentence.
Lines 168-169: Results from accelerometer readings show the more time older adults exhibited sedentary behaviors for an average of 10 hours per day,…
Please clarify this idea/sentence.
Lines 183-231:
There are several words bolded. Why?
Line 212:
”bridging the gap”
Font size is bigger.
Lines 225-231:
Please make clear that you are referring to your own previous article.
Line 267:
Sinergy?
Please add also a short mention about importance of other lifestyle factors (especially healthy plant-based diet, sleep and avoidance of risky substances like alcohol and tobacco).
Comments on the Quality of English LanguageThere are some phrases that could be written more clearly.
Author Response
Dear Editor and Reviewers,
We are grateful to you and the reviewers for the thorough evaluation of our manuscript entitled “Healthy behavior for preventing cognitive disability in older persons.” We appreciate the insightful feedback, which has helped us refine and improve our work significantly.
Below, we summarize the major revisions we have made in response to the reviewers’ comments (highlighted in red in the manuscript for ease of review):
- Clarity and Sentence Structure: We have revised specific sentences in the manuscript (noted in our point-by-point response) to ensure clarity, address grammatical concerns, and improve overall readability.
- Formatting and Style: We removed unintentional bold formatting and corrected any issues with font size to maintain consistency throughout the manuscript.
- Expansion on Lifestyle Factors: We added a short paragraph discussing the importance of a healthy diet (particularly plant-based), quality sleep, and avoidance of alcohol and tobacco as critical lifestyle components for older adults.
- Additional Content:
- We elaborated on why setting a moderate-to-vigorous physical activity target might be challenging for frail or comorbid older adults, emphasizing the practical relevance of light-intensity activity and minimized sedentary time.
- We introduced Table 1, summarizing key facilitators for successful and productive aging, and Table 2, which details the recommended physical activity guidelines, ensuring clearer visual presentation.
- Keywords: We have also reordered the keywords in alphabetical order, as requested.
We believe these changes align the manuscript more closely with the journal’s standards and enhance its contribution to geriatric research and clinical practice. We sincerely hope that our revisions address all concerns. Thank you again for the opportunity to improve our manuscript. We look forward to your response and remain available for any additional feedback or clarification.
Corresponding Author
Fulvio Lauretani, MD
Department of Medicine and Surgery, University of Parma, Parma, Italy
Geriatric Clinic, Geriatric-Rehabilitation Department, University of Parma, and University Hospital, Parma, Italy
Tel: +39-0521-703325
Email: fulvio.lauretani@unipr.it
Reviewer 2 Report
Comments and Suggestions for Authors
Dear authors,
Thank you for your contribution.
This is a well-written and informative paper. I have only few review comments, suggestions, and recommendations for your consideration as follows.
Abstract
Keywords: it is recommended to arrange keywords in an alphabetical order.
This paper is comprehensive and would have a positive significant impact on empowering older adults to maintain a healthy lifestyle and prevent cognitive decline.
In addition to what you have mentioned for the purposes of empowering older adults to maintain independence, and enhancing their overall wellbeing, I would suggest integrating other key strategies and helping factors, such as education for older adults and their caregivers/family members, building capacity and continuous training for healthcare providers to further support their journey of competency in geriatric care and provide high quality services per the sustainable development goal number three related to health and well-being, integrated model of care and the multidisciplinary teamwork to foster collaboration and integrated intervention planning, adoption of universal design principles and environmental modifications to support aging in place.
I recommend adding a table with key facilitative components for productive and successful aging. You already have this information in your text. However, I suggest putting it in a table to enhance readability and for a better visual representation.
6. Physical Activity: How Much Is Enough for Healthy Aging?
Again, to enhance readability and visual representation, I suggest putting the bullets underneath this point in a table.
I am happy to look at the revised manuscript.
Best wishes,
Author Response
Dear Editor and reviewers,
We are grateful to you and the reviewers for the thorough evaluation of our manuscript entitled “Healthy behavior for preventing cognitive disability in older persons.” We appreciate the insightful feedback, which has helped us refine and improve our work significantly.
Below, we summarize the major revisions we have made in response to the reviewers’ comments (highlighted in red in the manuscript for ease of review):
- Clarity and Sentence Structure: We have revised specific sentences in the manuscript (noted in our point-by-point response) to ensure clarity, address grammatical concerns, and improve overall readability.
- Formatting and Style: We removed unintentional bold formatting and corrected any issues with font size to maintain consistency throughout the manuscript.
- Expansion on Lifestyle Factors: We added a short paragraph discussing the importance of a healthy diet (particularly plant-based), quality sleep, and avoidance of alcohol and tobacco as critical lifestyle components for older adults.
- Additional Content:
- We elaborated on why setting a moderate-to-vigorous physical activity target might be challenging for frail or comorbid older adults, emphasizing the practical relevance of light-intensity activity and minimized sedentary time.
- We introduced Table 1, summarizing key facilitators for successful and productive aging, and Table 2, which details the recommended physical activity guidelines, ensuring clearer visual presentation.
- Keywords: We have also reordered the keywords in alphabetical order, as requested.
We believe these changes align the manuscript more closely with the journal’s standards and enhance its contribution to geriatric research and clinical practice. We sincerely hope that our revisions address all concerns. Thank you again for the opportunity to improve our manuscript. We look forward to your response and remain available for any additional feedback or clarification.
Corresponding Author
Fulvio Lauretani, MD
Department of Medicine and Surgery, University of Parma, Parma, Italy
Geriatric Clinic, Geriatric-Rehabilitation Department, University of Parma, and University Hospital, Parma, Italy
Tel: +39-0521-703325
Email: fulvio.lauretani@unipr.it